# Female Cardioprotection in a Mouse Model of Alcohol-Associated Cardiomyopathy

**DOI:** 10.3390/cells14211682

**Published:** 2025-10-27

**Authors:** Joshua M. Edavettal, Meagan Donovan, Nicholas R. Harris, Xavier R. Chapa-Dubocq, Keishla M. Rodríguez-Graciani, Janos Paloczi, Liz Simon, Bysani Chandrasekar, Jason D. Gardner

**Affiliations:** 1LSU Health Sciences Center, Department of Physiology, New Orleans, LA 70112, USA; jedav1@lsuhsc.edu (J.M.E.); mdono2@lsuhsc.edu (M.D.); nharr8@lsuhsc.edu (N.R.H.); xchapa@lsuhsc.edu (X.R.C.-D.); krodr7@lsuhsc.edu (K.M.R.-G.); jpaloc@lsuhsc.edu (J.P.); lsimo2@lsuhsc.edu (L.S.); 2Department of Medicine, School of Medicine, University of Missouri, Columbia, MO 65212, USA; chandrasekarb@health.missouri.edu; 3Harry S. Truman Veterans Memorial Hospital Columbia, MO 65212, USA; 4Dalton Cardiovascular Research Center, University of Missouri, Columbia, MO 65212, USA

**Keywords:** alcohol, chronic + binge, cardiovascular function, mitochondrial function, sex-differences, ACM, cardiomyopathy

## Abstract

Chronic alcohol misuse is the leading cause of non-ischemic dilated cardiomyopathy, and the molecular mechanisms underlying the development of alcohol-associated cardiomyopathy (ACM), particularly regarding sex-specific susceptibility and mitochondrial contributions, are not fully known. In this study, we utilized a preclinical model of chronic + binge ethanol consumption to investigate sex differences in disease severity and mitochondrial function. Male and female C57BL/6J mice were fed ethanol or control liquid diets for 30 days, with 2 binge episodes on days 10 and 30. Cardiac morphology was assessed via echocardiography and cardiac function via left ventricular pressure–volume catheterization. Mitochondrial function was evaluated ex vivo using Seahorse XF analysis, ATP luminescence, and AmplexTM Red fluorescence in isolated ventricular mitochondria. Ethanol feeding induced significant cardiac dysfunction and increased transcriptional expression of inflammatory and fibrotic markers in males, while these effects were not seen in females. Despite these sex-specific cardiac effects, mitochondrial respiration, ATP production, collagen protein expression, and oxidative stress were not significantly altered following alcohol exposure in either sex. Further investigation is warranted to assess the potential role of ovarian hormones in this female cardioprotection against chronic + binge ethanol.

## 1. Introduction

Excessive alcohol consumption remains a global health concern, ranking as one of the leading contributors to preventable death and disability worldwide [1]. Alcohol is linked to over 200 pathologies, including liver cirrhosis [2], pancreatitis [3], cancer [4], immune dysfunction [5], neurodegeneration [6], and cardiovascular dysfunction [7]. Cardiovascular disease (CVD) remains the leading cause of mortality worldwide and is responsible for over 20 million deaths annually [8]. While the effects of alcohol on hypertension [9], arrhythmogenesis [10], and direct myocardial toxicity are well-recognized, alcohol-associated cardiomyopathy (ACM) remains poorly studied. ACM represents the nexus of two major public health epidemics: excessive alcohol consumption and heart failure. This disease is marked by progressive ventricular dysfunction, structural remodeling, and a ten-year mortality rate of between 40–80% post-diagnosis [11]. Unfortunately, treatment options remain limited to symptomatic relief, abstinence, and, in rare cases, cardiac transplantation, as ACM is a diagnosis of exclusion and often recognized only in advanced stages.

Men are diagnosed more frequently than women with ACM, despite clinical data suggesting that women may be more vulnerable to ethanol-induced cardiac injury [11] and require less ethanol consumption to develop equivalent cardiac dysfunction [12]. These sex differences could be due to differences in metabolism and body composition, as females have lower gastric alcohol dehydrogenase (ADH) activity and total body water [13]. Decreased first-pass metabolism and volume of distribution lead to higher concentrations of alcohol and its toxic metabolite, acetaldehyde [13]. However, these are not the only sex differences related to cardiac outcomes, as estrogen increases vasodilation (decreasing strain on the heart), reduces cardiac hypertrophy, and reduces infarct sizes, while promoting recovery from ischemia–reperfusion injury [14]. Furthermore, estrogen signaling has a significant impact on bioenergetics; estrogen reduces oxidative stress and stabilizes mitochondrial activity [15]. Unfortunately, these protective effects decline with age, and their interaction with alcohol-mediated myocardial stress remains poorly investigated. However, differences in ethanol metabolism do not fully account for the sex distribution in ACM or its underlying pathophysiology. Therefore, preclinical investigations specifically focused on uncovering sex differences in ACM are needed.

One potential mechanism that may contribute to both alcohol-related cardiac injury and sex-based susceptibility is mitochondrial health. Mitochondrial dysfunction has been strongly implicated in the development and progression of heart failure, reflecting the heavy dependence of the heart on oxidative phosphorylation for ATP generation [16]. Impaired mitochondrial respiration, reduced ATP synthesis, and increased oxidative stress contribute to both systolic and diastolic dysfunction [17]. Reciprocally, heart failure itself exacerbates mitochondrial stress through increased energy demand, neurohormonal stimulation, and calcium dysregulation, creating a self-reinforcing cycle of bioenergetic dysregulation [18,19]. Chronic ethanol consumption impairs mitochondrial function and reduces ATP production [20,21]. While early studies established this connection [22], none have investigated mitochondrial function alongside direct measures of cardiac performance in experimental models of ACM. Despite known estrogenic effects on mitochondrial stability and oxidative balance [23,24], the potential for sex-specific differences in mitochondrial function remains largely unexplored.

To address the lack of study of sex-specific vulnerability in ACM, we investigated cardiac and mitochondrial function using a preclinical model of chronic ethanol consumption. We employed a previously established model using Lieber-DeCarli liquid diet in combination with chronic + binge alcohol administration, which consistently induces cardiac dysfunction [25,26,27]. Using this model, we assessed sex differences in cardiac function via echocardiography and left ventricular (LV) pressure–volume catheterization. Further, we evaluated LV expression of proinflammatory and profibrotic genes, and protein expression of fibrillar collagens I and III. To quantify mitochondrial function and oxidative stress, we conducted Seahorse XF analysis, ATP luminescence assays, and AmplexTM Red fluorescence using isolated ventricular mitochondria.

## 2. Materials and Methods

### 2.1. Animals

Male and female C57BL/6J mice (9–10 weeks old) were obtained from The Jackson Laboratory (Bar Harbor, ME) and allowed to acclimate for one week in the Louisiana State University Health Sciences Center (LSUHSC) animal care facility prior to experimental procedures. All mice were housed in a temperature- and humidity-controlled environment on a standard 12 h light/dark cycle (lights on from 6:00 a.m. to 6:00 p.m.; enrichment: better nest). During acclimation, mice were maintained on chow (Teklad 2019S, Envigo, Indianapolis, IN, USA) and provided autoclaved water ad libitum. All procedures were approved by the LSUHSC Institutional Animal Care and Use Committee and conducted in accordance with the NIH Guide for the Care and Use of Laboratory Animals. LSUHSC is accredited by the Association for Assessment and Accreditation of Laboratory Animal Care (AAALAC). Initial sample size was 4–6 per group, based on previous studies. After 4–6 echocardiographic samples were collected, a power analysis was performed to determine the sample size needed to reach significance. No mice were excluded from analyses.

### 2.2. Chronic + Binge Alcohol Administration

Alcohol-containing (F1258SP) and control (F1259) shake-and-pour liquid diets were purchased from Bio-Serv (Flemington, NJ, USA) and provided to the mice in specialized feeding tubes (9019 and 9015; Bio-Serv). Feeding tubes were cleaned daily, and fresh diet was provided. The general condition of the mice was observed during the daily diet change. All animals were housed two per cage. Mice were first acclimated to the control liquid diet for five days before random assignment to ethanol or control groups. The ethanol group (EtOH) received 5% (*v*/*v*) ethanol-containing diet for 30 days, while control animals were pair-fed an isocaloric control diet. Control mice were pair-fed based on the average alcohol consumption group’s consumption (12–15 mL/mouse/day). The 5% ethanol liquid diet produces midday blood alcohol levels (BAL) of 138 ± 83 and 115 ± 78 mg/dL for male and female mice, respectively (mean ± SD). Binge doses were administered as 5 g/kg alcohol in a 31.5% *v/v* alcohol-water solution via oral gavage. Peak BALs following ethanol gavage approach 400 mg/dL [25]. Control mice received an isocaloric gavage of maltose dextrin (9 g/kg; 45% *w*/*v*). Mice were gavaged between 12 and 2 PM.

### 2.3. Echocardiography

Cardiac structure and morphology were assessed via transthoracic echocardiography using a Vevo 3100 imaging system equipped with a 30 MHz probe (VisualSonics, Toronto, ON, Canada). Baseline and endpoint scans were acquired, with imaging performed at least 18 h after the last binge to avoid the acute cardiodepressive effects of the binge ethanol dosing. Mice were anesthetized with isoflurane (1.5%) and maintained at a heart rate between 400 and 500 bpm throughout the procedure. B- and M-mode images were captured in parasternal long-axis view and analyzed using Vevo Lab software (version 5, VisualSonics). Collection of echocardiography data and measurements was conducted by the same individual (not blinded) and performed using the leading-edge method across a minimum of three consecutive cardiac cycles and averaged for subsequent analysis.

### 2.4. LV Pressure–Volume (PV) Catheterization

Anesthesia was induced using 3–4% isoflurane and maintained at 2–3% after endotracheal intubation and mechanical ventilation. Mice were placed on a heated surgical platform, and body temperature was maintained with supplemental heat. A midline incision was made along the linea alba and extended through the sternum. Chest walls were retracted with metal hooks and secured. The apex of the heart was gently externalized using blunt forceps, and the pericardium was partially removed. A 25 G needle was used to puncture the apex and create an entry point, through which a PV catheter (SPR-839; Millar, Houston, TX, USA) was inserted into the LV chamber. Proper catheter placement was confirmed by real-time PV loop tracing. The investigator performing the heart catheterization was blinded to the treatment groups.

Steady-state, load-dependent PV measurements were recorded across multiple cardiac cycles. For load-independent assessments, the inferior vena cava (IVC) was transiently occluded using blunted forceps covered in silastic tubing. Resultant PV loop shifts were used to calculate end-systolic pressure–volume relationship (ESPVR; Ees) and preload-recruitable stroke work (PRSW). Parallel conductance was determined by injection of 0.1 mL hypertonic saline into the jugular vein. Volume calibration was performed using blood-filled cuvettes (Millar, 910-1049), and data were analyzed using LabChart 8 software (v8.1.24, ADInstruments). Data were analyzed by an investigator blinded to the treatment groups.

### 2.5. Tissue Collection

Following PV catheterization, hearts were removed and immediately placed in ice-cold saline to allow residual blood to be ejected, then gently blotted dry and weighed. The atria were removed, and the LV and interventricular septum (S) were separated from the right ventricle (RV). LV + S and RV weights were recorded and normalized to tibial length (TL). All weights and TL were measured by a blinded investigator.

### 2.6. Reverse Transcription Quantitative PCR (RT-qPCR)

Homogenization of LV tissue was performed with a Fisherbrand™ Pre-Filled Bead Mill Tube containing 1.4 mm ceramic beads (Cat. #15-340-153) using a Benchmark Scientific BeadBlaster™. RNA was extracted using the RNeasy Lipid Tissue Mini Kit (Qiagen, Cat. #74804). RNA concentration and purity were assessed via NanoDrop™ 1000 spectrophotometer (Thermo Scientific, Waltham, MA, USA), and only samples with A260/A280 ratios ≥ 1.8 were included for downstream analysis. Complementary DNA (cDNA) synthesis was performed using 2 µg of total RNA with the High-Capacity cDNA Reverse Transcription Kit (Applied Biosystems™, Cat. #4368814), following the thermal cycling protocol: 25 °C for 10 min, 37 °C for 120 min, and 85 °C for 5 min.

RT-qPCR was conducted using TaqMan Fast Advanced Master Mix (Thermo Fisher Scientific, Cat. #4444963) on a CFX Opus 96 Real-Time PCR System (BioRad, Hercules, CA, USA). TaqMan gene expression assays were used to quantify transcripts for Il1b (Mm00434228_m1), Il6 (Mm00446191_m1), Il10 (Mm01288386_m1), Larp6 (Mm00470891_m1), Col1a1 (Mm00801666_g1), and Col3a1 (Mm00802300_m1). PCR cycling conditions consisted of an initial activation at 95 °C for 20 s, followed by 40 cycles of 95 °C for 1 s and 60 °C for 20 s. Gene expression was calculated using the comparative CT method (ΔΔCT), with 18S rRNA (Mm03928990_g1) serving as the housekeeping control.

### 2.7. Western Blotting

Approximately 20–30 mg of LV tissue was homogenized in RIPA buffer (Millipore, Cat. #20-188) with HALT protease and phosphatase inhibitors (Thermo Scientific, Cat. #78441), and protein concentration was determined by BCA Protein Assay Kit (Thermo Scientific, Cat. #23227). Equal amounts of protein (15–20 µg) were mixed in Laemmli buffer (Bio Rad, Cat. #1610747) and 2-mercaptoethanol (Bio Rad, Cat. #1610710), resolved on 4–15% SDS-PAGE gels (Bio Rad, Cat. #4561086), and wet transferred to PVDF membranes overnight. Membranes were stained for total protein using No-Stain Protein Labeling Reagent (Invitrogen, Cat. #A44717), blocked with 5% (*w*/*v*) bovine serum albumin (Cell Signaling Technology, Cat. #9998) in Tris Buffered Saline with Tween 20 (TBS-T), and incubated overnight at 4 °C with primary antibodies against collagen I (1:1000, Abcam, AB34710), collagen III (1:1000, Abcam, AB7778), and α-smooth muscle actin (1:1000, Abcam, AB 5694). After washing, membranes were incubated with HRP-conjugated secondary antibodies (1:10,000, Cell Signaling Technology, Cat. #7074) for 1 h at room temperature. Bands were detected by chemiluminescence (Lumiglo, Cell Signaling Technology, Cat. #7003) and imaged with an Amersham Imager 680 (General Electric Company). Densitometric analyses were performed using the analysis program on the Amersham Imager 680 with normalization to total protein staining per lane. Protein expression was normalized relative to the control group.

### 2.8. Buffer Preparation for Seahorse Experiments

Two buffers were prepared for mitochondrial isolation and respirometry, as previously described [28]. Isolation buffer consisted of 70 mM sucrose, 210 mM mannitol, 5 mM HEPES (pH 7.2, adjusted with KOH), and 2 mM EGTA. To minimize proteolytic degradation during tissue processing, 0.5% bovine serum albumin (BSA) was included during the isolation step. Respiration buffer was composed of 10 mM sucrose, 220 mM mannitol, 2 mM HEPES (pH 7.4), 1 mM EGTA, 5 mM MgCl_2_, and 10 mM KH_2_PO_4_. Malate (2 mM) and pyruvate (10 mM) were added as Complex I substrates to initiate oxidative phosphorylation and support electron transport chain activity. All buffers were prepared fresh, filtered, and maintained on ice prior to use.

### 2.9. Mitochondrial Isolation

Mitochondria were isolated using a previously described protocol with minor modifications [29]. Hearts were harvested at least 24 h following the final ethanol binge. Mice were anesthetized with 4–5% isoflurane, and a thoracotomy was performed to expose the heart. Blood was removed from the heart via cardiac perfusion with chilled phosphate-buffered saline (PBS). Hearts were then excised and atria removed. Ventricular tissue was minced in 1–2 mL of ice-cold isolation buffer. This sample was kept on ice and mechanically disrupted using a tissue homogenizer for 10 s. The homogenizer was immediately rinsed with three 1 mL volumes of isolation buffer, which were pooled with the homogenate. The homogenate was transferred into 50 mL polycarbonate tubes (ThermoFisher Scientific, Cat. #3118-0050) pre-chilled on ice containing 15 mL of isolation buffer supplemented with 0.5% BSA to minimize proteolytic degradation. Samples were centrifuged at 2000× *g* for 4 min at 4 °C to pellet cellular debris. The supernatant was transferred to clean, pre-chilled 50 mL tubes and centrifuged at 10,000× *g* for 6 min at 4 °C to isolate crude mitochondria. Pellets were gently resuspended in 20 mL of fresh isolation buffer and centrifuged again at 10,000× *g* for 6 min at 4 °C to further purify the mitochondrial fraction. This resuspension and centrifugation step was repeated as needed to ensure purity. Final mitochondrial pellets were carefully resuspended in 60–100 µL of respiration buffer. After assays were performed, a 1:100 (*v*/*v*) dilution of protease inhibitor cocktail (ThermoFisher Scientific, Cat. #78444) was added to preserve mitochondrial integrity.

### 2.10. Bradford Assay for Protein Quantification

Protein concentrations of isolated mitochondrial samples were determined using the Bradford assay. Samples were diluted 1:3 by combining 4 µL of mitochondrial suspension with 8 µL of ultrapure deionized water. For each sample, triplicate wells were prepared by mixing 3 µL of diluted sample with 17 µL of water. BSA standards were prepared in ultrapure deionized water and aliquoted at 20 µL per well to generate a standard curve. For the assay, 1 mL of Bradford reagent (Cat. #22660, ThermoFisher Scientific, Waltham, MA, USA) was added to each sample and standard. A total of 200 µL of the mixture was transferred to a 96-well plate (Cat. #9017, Corning, Corning, NY, USA), and absorbance was measured at 595 nm using a Cytation C10 confocal imaging reader (Agilent). Data acquisition and analysis were performed using Gen5 Image software (version 3.14). Protein concentrations were calculated by comparing sample absorbance values to the BSA standard curve. All measurements were performed in triplicate and averaged.

### 2.11. Seahorse Assay for Mitochondrial Complex I-Dependent Respiration Analysis

Mitochondrial respiration was measured using the Seahorse XFe Pro Analyzer (Agilent, Santa Clara, CA, USA). Seahorse cartridges were hydrated overnight and calibrated using the manufacturer’s standard protocol prior to running the assay. Respiration buffer and injection reagents were freshly prepared on the day of the experiment to prevent pH drift or compound degradation. Freshly isolated mitochondria were loaded into each well at a concentration of 1.5 µg per well (total of 50 µL per well), followed by addition of 130 µL of respiration buffer, for a final assay volume of 180 µL. Assays began with a 12 min equilibration period, followed by sequential injections to induce distinct respiratory states:

Injection A: 4 mM ADP to initiate state 3 respiration.

Injection B: 2.5 µM oligomycin to inhibit ATP synthase, isolating proton leak (state 4o respiration).

Injection C: 4 µM FCCP to uncouple oxidative phosphorylation and measure maximal respiratory capacity (state 3µ respiration). FCCP concentration was optimized in preliminary titrations.

Injection D: 4 µM rotenone and 2 µM antimycin A to inhibit complexes I and III, respectively, quantifying non-mitochondrial oxygen consumption.

Each injection was followed by a 1 min mixing step and a 3 min measurement phase.

### 2.12. Seahorse XFe Pro Data Analysis

Oxygen consumption rate (OCR) values were corrected by subtracting non-mitochondrial OCR. The non-mitochondrial OCR was determined following the injection of rotenone and antimycin A. Wells that failed to respond to ADP or FCCP were excluded from analysis to ensure accuracy and consistency across experimental groups. Mitochondrial respiration parameters were extracted from Seahorse Wave software (version 10.1.0). These values were used to assess various states of respiration—basal (State 2), ADP-stimulated (State 3), non-productive (State 4o), and uncoupled (State 3µ).

### 2.13. ATP Quantification via Luminescence Assay

ATP content was measured using the CellTiter-Glo^®^ 2.0 Luminescent Cell Viability Assay (Cat. #G9241, Promega, Madison, WI, USA) according to the manufacturer’s protocol. Samples and standards were prepared in an opaque 96-well plate (Cat. #9017, Corning, Corning, NY, USA). ATP standards were prepared from a 10 mM stock solution and serially diluted in assay buffer to generate a standard curve ranging from 0 to 50 µM. Each standard was plated in duplicate (40 µL/well), and mitochondrial samples were plated in triplicate at a concentration of 50 µg in 40 µL per well. CellTiter-Glo^®^ 2.0 reagent was added at a 1:1 ratio (40 µL per well), and plates were shaken on an orbital shaker for 5 min, then incubated at room temperature for 25 min to stabilize the luminescent signal. Luminescence was recorded using a Cytation C10 plate reader (Agilent), and ATP concentrations were calculated from the standard curve using Gen5 Image software. ATP content values were normalized to protein content.

### 2.14. Amplex Red Assay for Hydrogen Peroxide Quantification

Hydrogen peroxide (H_2_O_2_) production was measured using the Amplex^®^ Red Hydrogen Peroxide/Peroxidase Assay Kit (ThermoFisher Scientific), following the manufacturer’s protocol. Fluorescence was recorded using a microplate reader (Cytation C10, Agilent) with excitation at 540 nm and emission at 590 nm. Freshly isolated mitochondria were diluted in 1X Reaction Buffer to a final concentration of 0.5 µg/µL. A 50 µL aliquot of each mitochondrial sample (in triplicate) was added to individual wells. H_2_O_2_ standards (0–50 µM) were prepared in 1X Reaction Buffer and plated at 50 µL per well in duplicate. The working solution was prepared immediately before use and consisted of 100 µM Amplex^®^ Red reagent and 0.2 U/mL horseradish peroxidase (HRP), prepared by mixing 50 µL of 10 mM Amplex Red stock, 100 µL of 10 U/mL HRP stock, and 4.85 mL of 1X Reaction Buffer. A total of 50 µL of working solution was added to each well, yielding a final volume of 100 µL. Plates were incubated for 30 min at room temperature, protected from light. Fluorescence was measured kinetically at 5 min intervals for one hr. Background signal was subtracted using the 0 µM H_2_O_2_ control, and end-point fluorescence values were converted to H_2_O_2_ concentrations based on the standard curve.

### 2.15. Statistical Analyses

Data are presented as mean ± standard error of the mean (SEM). For this 2 × 2 design, statistical comparisons were conducted using two-way ANOVA (sex and ethanol variables). Tukey’s post hoc analyses were performed to assess differences between group means. Data were evaluated for normality prior to ANOVA. All statistical tests were conducted using Prism version 10 (GraphPad Software, Boston, MA, USA), and significance was defined as *p* < 0.05.

## 3. Results

### 3.1. Heart Weights and Echocardiographic Findings Were Minimal in Ethanol-Fed Male and Female Mice

To assess the impact of chronic + binge alcohol consumption on cardiac size, hearts were collected after LV PV catheterization, dissected, and weighed. Total heart weight was normalized to tibial length (TL) to account for body weight differences. No significant differences in total heart weight/TL ratios were observed between ethanol-fed and control females (Figure 1A), while ethanol-fed male mice exhibited a trend for lower heart weight (male cohort was previously reported [27]). To further assess chamber-specific changes, the atria were removed, and the LV and interventricular septum (LV + S) were separated from the RV. Neither LV + S (Figure 1B) nor RV (Figure 1C) weight differed between ethanol- and control-treated females, while there was a trend for lower LV + S mass in ethanol-fed males. Body weight was also not significantly different in males or females (Figure 1D) fed ethanol, although males did show indications of lower body weights. Tibial length did not differ across any groups (Figure 1E).

Echocardiography was performed prior to catheterization to noninvasively assess cardiac morphology. Overall, few ethanol effects or sex differences were detected. Interestingly, although no differences were observed in cardiac mass, LV internal diameter in diastole (LVID;d) was significantly reduced in ethanol-fed females (Figure 2B). Both LV anterior wall thickness and posterior wall thickness at diastole (LVAW;d and LVPW;d) indicated an interaction by 2-way ANOVA (Figure 2A,C). There were no significant changes noted for systolic chamber and wall dimensions. There were no significant changes in either fractional shortening or ejection fraction between groups (See Appendix A).

### 3.2. Left-Ventricular PV Catheterization Revealed Preservation of Cardiac Function in Ethanol-Fed Female Mice

To directly assess in vivo cardiac performance, we performed terminal LV PV catheterization. Ethanol-fed female mice exhibited preserved systolic and diastolic function, showing no significant impairments across key cardiac functional parameters. In contrast, ethanol-fed males demonstrated marked systolic dysfunction, including reduced maximum (Max) pressure (Figure 3A) and decreased contractility, as measured by dP/dt Max (Figure 3C). 2-way ANOVA indicated interaction between Sex and Ethanol for these parameters, and for the end-systolic pressure–volume relationship (ESPVR), a load-independent index of intrinsic contractile function (Figure 3D). Diastolic function was also impaired in males, with reduced dP/dt Min (Figure 3E) and a trend toward prolonged ventricular relaxation time constant, Tau (Figure 3F). For all measures, males exhibited greater systolic and diastolic dysfunction than females in response to ethanol.

### 3.3. Female Mice Were Protected from Ethanol-Induced Upregulation of Proinflammatory and Profibrotic Genes

LV tissue was collected for RT-qPCR analysis to assess inflammatory and fibrotic gene expression. After 30 days, ethanol-fed male mice had increased expression levels of Il6 and Il1b relative to control males, with no significant changes in Traf3ip2 or Il10. In contrast, ethanol-fed female mice exhibited no change in expression levels of Il6 or Il1b (Figure 4A–D).

Similarly, profibrotic markers, Col1a1, Col3a1 and Larp6, were not altered in the LV of ethanol-fed females. (Figure 5A–D). In contrast, ethanol-fed male mice exhibited a significant reduction in Col3a1 expression, resulting in an increase in Col1a1/Col3a1 expression ratio. Male mice also had increased expression levels of Larp6 in response to ethanol feeding.

### 3.4. Protein Expression of Profibrotic Markers Was Not Significantly Increased in Response to Ethanol Feeding in Either Sex

Assessment of protein expression by Western blotting indicated no interaction, and no main effect of Sex or Ethanol on LV expression of collagen I, collagen III, or α-smooth muscle actin (Figure 6).

### 3.5. Mitochondrial Respiration Was Higher in Male Mice than in Female Mice, but Was Not Significantly Affected by Chronic + Binge Ethanol Feeding

Mitochondrial respiration was assessed via Seahorse XF analysis in isolated ventricular mitochondria to evaluate bioenergetic function. OCR was measured across distinct respiratory states to quantify substrate-driven, ADP-stimulated, non-productive, and uncoupled respiration. No significant differences in OCR were observed between ethanol-fed and control groups in either sex. However, mitochondria isolated from male hearts consistently exhibited higher OCR values across multiple states of respiration, including basal (State 2; Figure 7A), ADP-stimulated (State 3; Figure 7B), and uncoupled respiration (State 3µ; Figure 7D), compared to females. State 4o respiration, representing proton leak following oligomycin treatment, was not significantly different between groups (Figure 7C).

### 3.6. ATP Production and Hydrogen Peroxide Generation Were Not Significantly Altered by Ethanol Feeding or Sex

To assess the downstream functional consequences of mitochondrial respiration, ATP production and H_2_O_2_ generation were measured in isolated ventricular mitochondria. Across all groups, there were no significant differences in ATP levels or H_2_O_2_ generation (Figure 8A,B). Neither sex nor ethanol feeding significantly impacted these measures.

## 4. Discussion

The goal of this study was to examine sex differences in ACM, an area that remains underexplored in both clinical and preclinical research. Our findings demonstrate striking protection against the pathological effects of chronic + binge ethanol feeding in female C57BL/6J mice. Female mice exhibited no significant impairments in cardiac function, in contrast to males, which exhibited significant systolic and diastolic dysfunction. Males also developed increased expression of proinflammatory and profibrotic genes in response to chronic + binge ethanol feeding, where females did not. Although male mice showed higher mitochondrial OCR compared to females, alcohol exposure did not significantly impair mitochondrial respiration and ATP production or increase hydrogen peroxide generation in either sex. These results highlight the divergence in how males and females respond to chronic + binge ethanol consumption and suggest that sex-specific protective mechanisms may mitigate the development of ACM in females.

In male mice, LV mass showed a decreasing trend following chronic + binge ethanol, which may indicate early pathological remodeling [27,30,31]. Female mice exhibited no changes in total heart, LV or RV mass. The preservation of cardiac structure in females may contribute to their protection against the functional impairments observed in males. These findings align with prior work in Sprague-Dawley rats, where chronic ethanol exposure over 26 wks resulted in reduced heart weight and thinner ventricular walls in males but not females [32]. Tibial length did not differ across our groups, indicating that the observed cardiac differences were independent of body size variation (Figure 1).

The echocardiography findings were not remarkable, with very few differences between control and ethanol-fed mice. In contrast, LV PV catheterization revealed that male mice developed significant systolic and diastolic dysfunction in response to chronic + binge ethanol feeding. Surprisingly, the cardiac function of female mice was largely unaffected by chronic + binge ethanol, with females exhibiting no significant changes across both systolic and diastolic functional measurements. Ethanol-fed female mice maintained normal cardiac function, including maximum pressure generation, cardiac stroke work, dP/dt Max, and ESPVR, indicating preserved systolic function and intrinsic myocardial contractility. Diastolic performance was similarly unaffected by ethanol in females, with no significant changes in dP/dt Min or Tau. In contrast, ethanol-fed male mice exhibited significant impairments in these diastolic parameters, consistent with the development of ACM. These findings highlight a sex-specific resilience to chronic ethanol-induced myocardial injury, with female mice exhibiting cardioprotection from the functional deficits observed in males.

The preservation of cardiac function in female mice was further supported at the molecular level by the absence of elevation in proinflammatory genes following chronic + binge ethanol feeding. In male mice, LV expression of the pro-inflammatory cytokines Il6 and Il1β was significantly increased relative to controls, while Traf3ip2 expression was unchanged. The expression of the anti-inflammatory cytokine Il10 also remained unchanged. These findings align with previous reports that chronic and excessive alcohol consumption upregulates myocardial inflammatory pathways, promoting the progression of heart failure [33,34]. Low and moderate levels of alcohol intake have been associated with reductions in systemic inflammatory markers [35], and epidemiological studies suggest that males exhibit a more linear increase in inflammatory markers with ethanol consumption, whereas females demonstrate a J-shaped association, with low amounts potentially exerting anti-inflammatory effects [36]. However, evidence has also challenged the notion of cardioprotective alcohol effects in females, instead emphasizing that even lower-volume ethanol intake is harmful [10]. Despite these complexities, there remains a paucity of preclinical and clinical comparative studies examining myocardial responses to chronic ethanol exposure between sexes.

Analysis of fibrotic markers further emphasized the divergence in molecular responses between sexes following chronic + binge ethanol feeding, with males exhibiting a profibrotic transcriptional shift in response to ethanol. Female mice showed no significant changes in Col1a1, Col3a1, or Larp6 mRNA expression, consistent with their preserved diastolic function. The absence of molecular evidence for fibrotic remodeling in females further supports the hypothesis that intrinsic sex-specific mechanisms confer protection against ethanol-induced myocardial damage. However, in male mice, we observed a significant decrease in Col3a1 expression and a corresponding increase in the Col1a1/Col3a1 ratio, indicative of a shift in extracellular matrix composition toward stiffer, less compliant myocardium. This remodeling pattern is consistent with fibrosis, as increased collagen type I relative to type III reduces ventricular compliance and promotes stiffness [37,38]. Additionally, males exhibited significantly elevated expression of Larp6, a regulator of collagen type I mRNA stability [39], suggesting activation of profibrotic pathways at the post-transcriptional level. However, these changes in mRNA expression were not reflected at the protein level, with neither male nor female mice developing significant changes in collagen I or III expression in response to ethanol. We previously found that histological collagen staining by picrosirius red was not increased in male mice fed chronic + binge ethanol despite significant changes in mRNA expression [27]. Although we exposed mice to 30 d of chronic + binge ethanol, longer durations may be needed to realize changes in cardiac collagen protein expression.

One potential mechanism underlying the observed sex differences in cardiac vulnerability is mitochondrial function. Mitochondrial oxidative phosphorylation is the primary source of ATP generation in the heart [40], and both contraction and relaxation are critically dependent on adequate mitochondrial energy production. Consequently, mitochondrial dysfunction is a major contributor to the development and progression of heart failure. Increased energetic demands during cardiac stress further exacerbate mitochondrial injury, creating a cycle of bioenergetic collapse [41]. Given that chronic + binge ethanol feeding imposes substantial hemodynamic and metabolic stress on the heart, mitochondrial dysfunction may contribute to the systolic and diastolic impairments observed in the ethanol-fed male mice. Sex-specific differences in mitochondrial biology have been documented in preclinical models, with several studies suggesting that females possess inherently greater mitochondrial resilience. Mitochondrial efficiency, glutamate/malate-stimulated respiration, and fatty acid oxidation during exercise are enhanced in females, while mitochondrial content, ROS production, and calcium uptake rates are higher in males [42]. These differences are thought to be due to increased mitochondrial biogenesis and the antioxidant properties of estrogen [43]. A genome-wide expression profiling study in rat hearts demonstrated lower expression of apoptotic genes and upregulation of oxidative phosphorylation-related transcripts in aged females compared to males [44]. These findings suggest that intrinsic sex differences in mitochondrial stability and bioenergetics may contribute to the cardioprotection observed in female mice following chronic + binge ethanol feeding.

Mitochondrial function was evaluated using Seahorse XF analysis of isolated ex vivo ventricular mitochondria, providing real-time assessment of oxygen consumption rates across key respiratory states. Cardiac mitochondria from males exhibited significantly higher oxygen consumption compared to mitochondria from females, consistent with previous reports of intrinsic sex differences in cardiac bioenergetics [45]. Specifically, mitochondria from males exhibited elevated basal respiration (State 2), ADP-stimulated respiration (State 3), and maximal uncoupled respiration (State 3µ), whereas proton leak–associated respiration (State 4o) did not differ between sexes. These findings suggest that male mice possess higher intrinsic mitochondrial oxidative capacity under both baseline and stimulated conditions, but do not exhibit differences in non-productive respiration. Interestingly, despite these baseline sex differences in mitochondrial respiration, chronic + binge ethanol feeding did not significantly impair mitochondrial oxygen consumption in either sex. This finding suggests that while mitochondrial oxidative capacity differs between males and females, ethanol-induced cardiac dysfunction in our ACM model was not associated with overt mitochondrial Complex I-dependent respiratory dysfunction.

Mitochondrial ATP production and H_2_O_2_ generation were also assessed to evaluate downstream functional consequences of chronic + binge ethanol consumption. We found no significant differences in ATP content or H_2_O_2_ between sexes or treatment groups. This contrasts with prior literature showing that ethanol disrupts mitochondrial bioenergetics, leading to reduced ATP production and increased oxidative stress [20,21,46]. However, most studies have relied on in vitro models with direct ethanol exposure, whereas our ex vivo measurements utilized freshly isolated ventricular mitochondria from ethanol-fed mice. Another potential difference is that cardiac mitochondria demonstrate high intrinsic respiratory efficiency compared to other muscle types, which may allow them to maintain ATP output even under pathological stress. The absence of ethanol-induced reductions in ATP production or elevations in H_2_O_2_ suggests that at this stage of exposure, mitochondrial energetic function and antioxidant defenses remain largely intact. This is consistent with the preserved mitochondrial respiration observed in Seahorse assays and may explain, in part, why overt energetic collapse was not detected despite clear ventricular functional impairments in the alcohol-fed males. The relatively low alcohol dehydrogenase activity in cardiac tissue, compared to the liver, may also limit local ROS production during alcohol metabolism, reducing mitochondrial oxidative burden. Together, these data indicate that mitochondrial dysfunction and oxidative stress may not be the primary drivers of early cardiac impairment in this ACM model.

While this study provides important insights into sex-specific differences in ACM, some limitations must be acknowledged. Although the chronic + binge alcohol exposure model recapitulated key aspects of early-stage ACM, it does not fully replicate the prolonged, multifactorial nature of human disease development. This chronic + binge ethanol feeding mouse model is one of the few ethanol exposure models that produces significant cardiomyopathy. However, the model does not fully capture the severe ventricular fibrosis that is a clinical hallmark of ACM. This difference could be due to the relatively short ethanol exposure period, which may capture early pathological changes but not the extensive fibrosis or end-stage remodeling observed clinically. Mitochondrial function was assessed ex vivo, which allowed for precise measurements of respiratory parameters but may not fully capture in vivo metabolic interactions or time-dependent adaptations. The Seahorse approach used in this study evaluates Complex I-dependent respiration [28]; however, the addition of succinate would allow assessment of Complex II-linked respiration, and the inclusion of palmitoyl-carnitine/ADP would provide valuable information regarding fatty acid oxidation capacity. Other important mitochondrial functional aspects that could be involved are mitochondrial dynamics (i.e., fission and fusion). Furthermore, although female mice exhibited striking cardioprotection, we did not synchronize or time the estrous cycle of the mice or directly manipulate or measure sex hormone levels. Further studies are warranted to address the role of estrogen, progesterone or other ovarian hormones in the cardioprotective effects demonstrated.

Future studies incorporating additional longitudinal assessments of cardiac function, hormonal interventions, and broader mechanistic analyses are needed to establish the molecular basis of ethanol-induced cardiac dysfunction. Further, extending the ethanol exposure duration or incorporating additional cardiovascular stressors to promote a fibrotic response could clarify the long-term impact of ethanol on cardiac remodeling and may more closely approximate the human disease phenotype of ACM.

## 5. Conclusions

This study reveals significant sex-specific differences in cardiac vulnerability to chronic + binge ethanol feeding, with male mice exhibiting cardiac functional impairment, and a molecular transcriptional profile suggestive of inflammatory and fibrotic activation, while female mice were protected across all measured parameters. Despite baseline differences in mitochondrial respiration between sexes, ethanol did not significantly impair mitochondrial oxidative phosphorylation, ATP production, or increase ROS generation in either sex. These findings advance our understanding of the adverse effects of ethanol on the heart and provide evidence of sex-specific cardioprotection in the chronic + binge ethanol mouse model of ACM.

## Figures and Tables

**Figure 1 cells-14-01682-f001:**
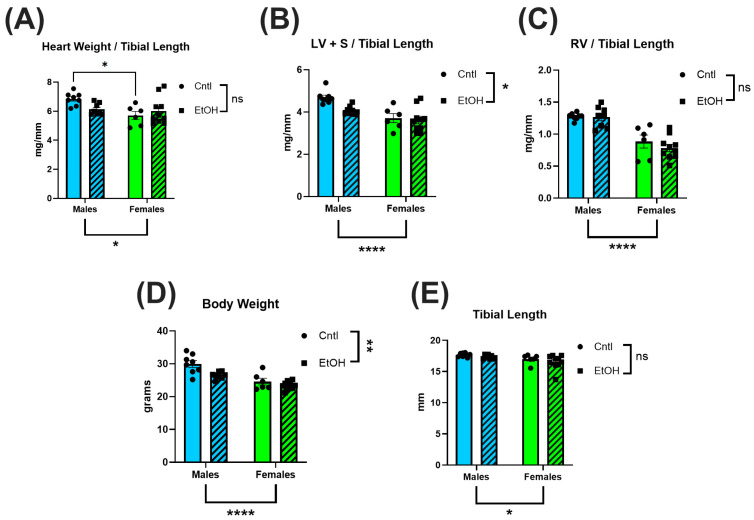
Chronic + binge ethanol feeding did not significantly impact heart or body weights in male or female mice after 30 d. Total heart weight (**A**), LV and interventricular septum weight (LV + S) (**B**), and RV weight (**C**) were normalized to tibial length. Body weights (**D**) were lower overall in males and in the ethanol-fed groups. No differences were found in tibial length (**E**). Sample size per group: *n* = 6–8. Control (Cntl) animals are shown as circles with solid bars; ethanol-fed (EtOH) animals are shown as squares with dashed bars. These data were analyzed using a two-way ANOVA followed by Tukey’s post hoc test for group comparisons. Main effects are denoted below the *y*-axis (Sex effect) and on the group legend (Ethanol effect). ns = *p* > 0.05, * *p* < 0.05, ** *p* < 0.01, **** *p* < 0.0001.

**Figure 2 cells-14-01682-f002:**
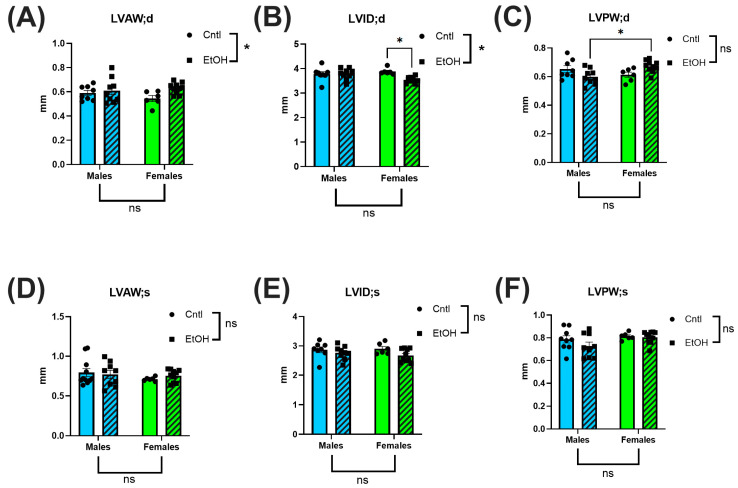
Echocardiography showed minimal morphological changes following chronic + binge ethanol feeding. Left ventricular anterior wall (LVAW), internal diameter (LVID), and posterior wall (LVPW) (in both diastole (d) and systole (s)) were assessed. Sample size per group: *n* = 6–8. Control (Cntl) animals are shown as circles with solid bars; ethanol-fed (EtOH) animals are shown as squares with dashed bars. These data were analyzed using a two-way ANOVA followed by Tukey’s post hoc test for group comparisons. ns = *p* > 0.05, * *p* < 0.05.

**Figure 3 cells-14-01682-f003:**
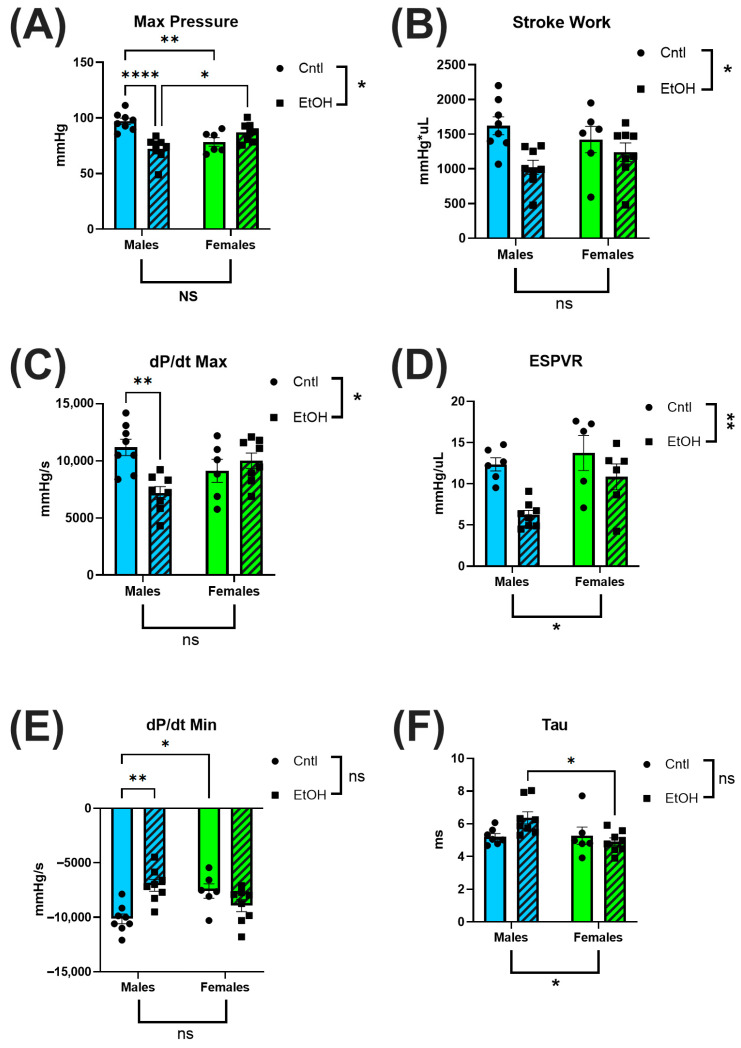
Left ventricular pressure–volume catheterization revealed sex-specific cardiac dysfunction following chronic + binge ethanol feeding after 30 days. Ethanol-fed females did not differ from controls in systolic or diastolic parameters, whereas ethanol-fed males showed significant impairments. Maximum pressure (**A**), stroke work (**B**), and dP/dt Max (**C**) were all significantly reduced in ethanol-fed males, indicating compromised systolic function. The end-systolic pressure–volume relationship (ESPVR; (**D**)) did not indicate a significant interaction between Sex and Ethanol, but was greatly reduced in ethanol-fed males. Diastolic dysfunction was reflected by reduced dP/dt Min (**E**) and a trend toward prolonged relaxation time constant, Tau (**F**), only in ethanol-fed males. Sample size: *n* = 5–8 in each group. Control (Cntl) animals are shown as circles with solid bars; ethanol-fed (EtOH) animals are shown as squares with dashed bars. These data were analyzed using a two-way ANOVA followed by Tukey’s post hoc test for group comparisons. ns = *p* > 0.05, * *p* < 0.05, ** *p* < 0.01, **** *p* < 0.0001.

**Figure 4 cells-14-01682-f004:**
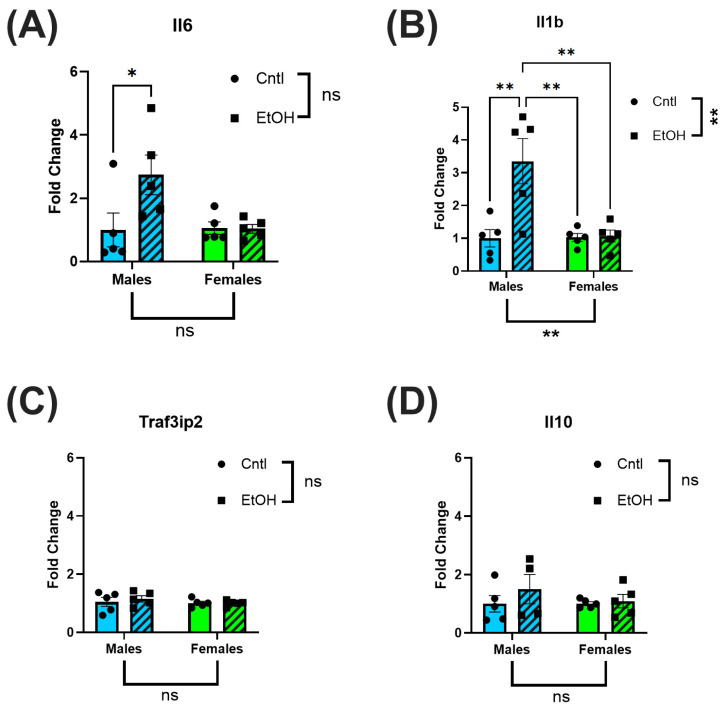
Ethanol-fed female mice did not exhibit increases in inflammatory markers. RT-qPCR analysis of inflammatory biomarkers in LV homogenates of ethanol-fed mice indicated elevated Il6 (**A**) and Il1b (**B**) in male but not in female mice; Traf3ip2 (**C**) and Il10 (**D**) were not significantly different across all groups. These data were analyzed using two-way ANOVA followed by Tukey’s post hoc test for group comparisons. *n* = 5 in all groups. ns = *p* > 0.05, * *p* < 0.05, ** *p* < 0.01.

**Figure 5 cells-14-01682-f005:**
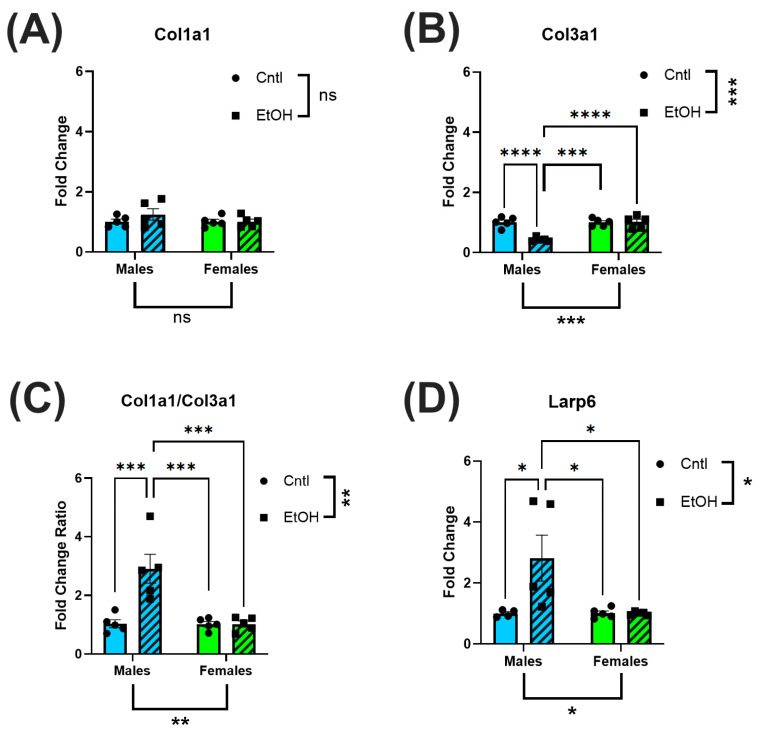
The LV of ethanol-fed female mice did not exhibit increases in fibrotic markers in response to chronic + binge ethanol feeding. RT-qPCR assessment of LV homogenate for fibrotic biomarkers indicated that while Col1a1 levels were not significantly altered (**A**), there was a decrease in Col3a1 (**B**) in male mice, resulting in an increase in the Col1a1/Col3a1 ratio (**C**); further, there was an increase in Larp6 (**D**). None of these changes were present in ethanol-fed female mice. These data were analyzed using a two-way ANOVA followed by Tukey’s post hoc test for group comparisons. *n* = 5 in all groups. ns = *p* > 0.05, * *p* < 0.05, ** *p* < 0.01, *** *p* < 0.001, **** *p* < 0.0001.

**Figure 6 cells-14-01682-f006:**
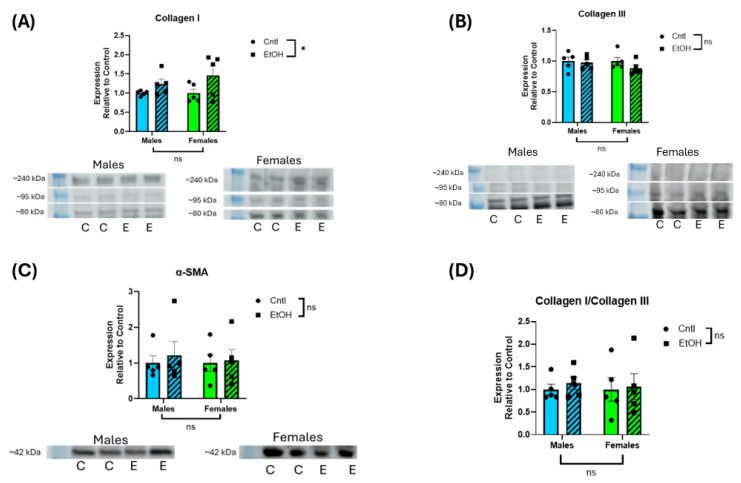
Ethanol-fed male and female mice did not exhibit increases in LV protein expression of fibrotic markers. Western blot assessment of LV homogenate indicated that collagens I (**A**) and III (**B**), α-smooth muscle actin (SMA) (**C**), and collagen I/III ratio (**D**) did not significantly change following chronic + binge ethanol feeding in either male or female mice. For collagen I and III, the three prominent bands between ~80 and 240 kDa were combined for quantification. Results were normalized to total protein of the lane. These data were analyzed using a two-way ANOVA followed by Tukey’s post hoc test for group comparisons. *n* = 5 in all groups. ns = *p* > 0.05, * *p* < 0.05. Collagen I indicated a main effect of ethanol, but there were no group differences in the post hoc analyses (C = control; E = ethanol). Uncropped blots, total protein stain, and quantification of individual collagen I and II bands are provided in the Appendix A.

**Figure 7 cells-14-01682-f007:**
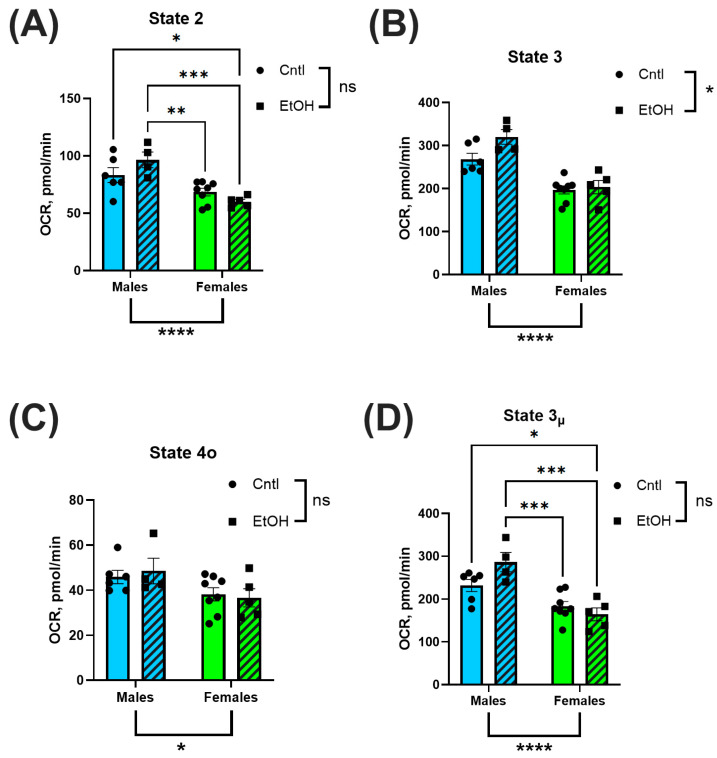
Mitochondrial Complex I-dependent respiration was not significantly altered by chronic + binge ethanol feeding but did vary by sex. Oxygen consumption rate (OCR) was assessed using Seahorse XF analysis in isolated ventricular mitochondria. Male mice exhibited significantly higher OCR across basal (State 2; (**A**)), ADP-stimulated (State 3; (**B**)), and uncoupled (State 3µ; (**D**)) respiration compared to females. There were no significant sex effects observed for proton leak (State 4o; (**C**)). There were no main effects of Ethanol for any of these mitochondrial respiration parameters. Data were analyzed using two-way ANOVA with Tukey’s post hoc test. Sample sizes: Cntl male *n* = 6, EtOH male *n* = 4, Cntl female *n* = 8, EtOH female *n* = 5. ns = *p* > 0.05, * *p* < 0.05, ** *p* < 0.01, *** *p* < 0.001, **** *p* < 0.0001.

**Figure 8 cells-14-01682-f008:**
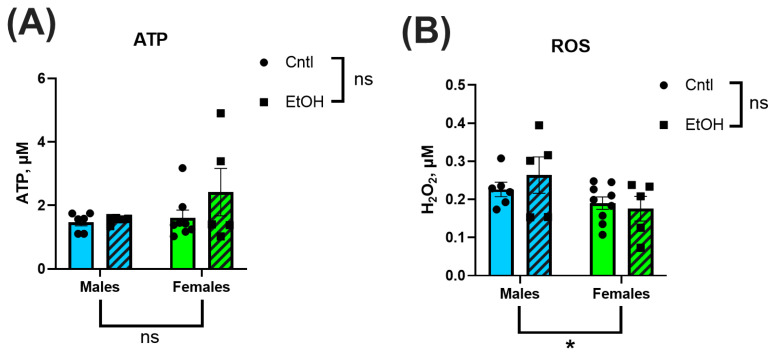
ATP generation (**A**) and reactive oxygen species (ROS) production (**B**) were measured in isolated ventricular mitochondria using luminescence- and fluorescence-based assays, respectively. There were no significant Sex or Ethanol effects in ATP or ROS production. Data were analyzed using two-way ANOVA followed by Tukey’s post hoc test. Sample sizes: Cntl male *n* = 6, EtOH male *n* = 4, Cntl female *n* = 8, EtOH female *n* = 5. H_2_O_2 =_ hydrogen peroxide. ns = *p* > 0.05, * *p* < 0.05, See Appendix A for additional information.

## Data Availability

See Appendix A for additional data and details of the statistical results. All data are available upon request. Please contact the corresponding author.

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
