# Peer review of "Female Cardioprotection in a Mouse Model of Alcohol-Associated Cardiomyopathy"

_cells, 2025, doi:10.3390/cells14211682_

Round 1

Reviewer 1 Report

Comments and Suggestions for Authors

The authors evaluate sex differences in a chronic-plus-binge ethanol exposure model in mice. They report that females exhibit relative protection from alcohol-induced cardiac dysfunction and transcriptional remodeling observed in males. Mitochondrial respiration (OCR), ATP levels, and H₂O₂ production measured in isolated left-ventricular mitochondria are reported as unchanged by ethanol at the sampled timepoint; baseline OCR appears higher in males. The manuscript concludes that female cardioprotection is independent of mitochondrial sex differences.

Major Comments

  • Causal claim exceeds evidence (conclusions/title).
    The data show (a) sex differences at baseline (e.g., higher OCR in males) and (b) no detectable treatment-induced change in selected mitochondrial readouts at a single timepoint. This does not establish that female cardioprotection is independent of mitochondrial mechanisms. Please temper the claim and title accordingly or add targeted perturbation experiments (e.g., ovariectomy/estrogen modulation; mitochondrial pathway inhibition/activation; genetic or pharmacologic manipulation)

  • Model timing and acute-effect confounding.
    The timing of echocardiography and hemodynamics relative to the last binge must be specified precisely and consistently. Small inconsistencies between Methods and figure legends (e.g., ≥12 h vs 18–24 h) materially affect interpretation because acute ethanol can transiently alter function. Provide exact intervals for each cohort, ensure consistency across the manuscript, and (if necessary) re-analyze excluding animals within acute windows.

  • Anesthesia, operator blinding, and bias control.
    Echocardiography under isoflurane without explicit blinding can bias functional readouts. Please report: (i) whether image acquisition and analysis were blinded; (ii) isoflurane concentration/timing; (iii) whether depth was matched across groups; and (iv) inter- and intra-operator variability or QC checks.

  • Statistical analysis inflates type-I error.
    Several figures appear to use two-way ANOVA followed by multiple post-hoc tests and additional groupwise t-tests highlighted on plots. This “double testing” increases false positives. Pre-specify a single analysis path, control FDR across families of related endpoints, and report effect sizes with 95% CIs.

  • Ambiguity in Seahorse data processing and QC.
    The Methods describe OCR values being “normalized to non-mitochondrial respiration” and wells with inadequate responses to ADP or FCCP being excluded. Standard practice is subtraction of non-mitochondrial OCR rather than ratio scaling.

  • Validation gaps for H₂O₂ assays on isolated mitochondria.H₂O₂: Amplex Red assays should include catalase controls (to verify signal identity), and presentation of substrate-dependent rates (e.g., pyruvate/malate vs palmitoylcarnitine) with/without SOD to confirm matrix origin. Please add these controls, raw calibration curves, and representative traces.

  • Model characterization is incomplete (BAC and systemic exposure).
    Blood alcohol concentrations (BAC) at defined timepoints—especially around imaging and tissue harvest—are essential to distinguish acute vs cumulative effects and to benchmark the model against the literature. Please provide BAC (or breath levels) and body-weight/consumption trajectories per sex.

  • Female hormonal status not controlled/reported.
    Given the sex-difference emphasis and estrogen’s known cardioprotective roles, estrous staging or hormone profiling is needed. At minimum, report cycle staging at testing or provide ovariectomy ± estradiol replacement data to test estrogen dependence.
  • ECM/fibrosis claims need orthogonal validation.
    Inferences about remodeling derived from mRNA (e.g., Col1a1/Col3a1 ratios, LARP6) should be corroborated at the protein and tissue level (Picrosirius Red or Masson trichrome, hydroxyproline, collagen I/III and TGF-β signaling proteins, or biaxial mechanics if stiffness is claimed).

Minor revision

  • Standardize nomenclature (e.g., “EtOH,” “HFrEF” if referenced, gene and protein formatting).

  • Verify reference formatting and ensure all in-text claims are supported by current literature.

  • Improve figure readability: ensure axis labels include units, avoid overplotting, and present mean ± SD/SEM consistently with superimposed datapoints.

  • Correct typographical errors and ensure consistent use of symbols (e.g., μ vs u).

Author Response

We appreciate the comments from the reviewers which have led us to include additional data and edits to many sections of the manuscript.  We believe that these edits have greatly improved the clarity and overall quality of the manuscript.

  • Causal claim exceeds evidence (conclusions/title).
    Please temper the claim and title accordingly or add targeted perturbation experiments.

We agree with the reviewer that aspects of our conclusions were overstated, particularly in regard to the limited mitochondrial assays that were performed which do not assess all aspects of mitochondrial function.  As such, we have changed the title and have expanded the discussion of limitations and future avenues of investigation (Last two paragraphs of the Discussion section)

  • Model timing and acute-effect confounding.
    The timing of echocardiography and hemodynamics relative to the last binge must be specified precisely and consistently. Small inconsistencies between Methods and figure legends (e.g., ≥12 h vs 18–24 h) materially affect interpretation because acute ethanol can transiently alter function. Provide exact intervals for each cohort, ensure consistency across the manuscript.

This is an important clarification as ethanol intoxication can have a dramatic effect on cardiovascular function. We have updated the Methods and Figure Legends to consistently report when we performed the cardiac functional assessments, which should allow other researchers to replicate our experimental conditions.  The echocardiography and cardiac catheterization were performed at least 18 hr after the alcohol binge dosing.  Even though the mice still have access to the liquid diet throughout the day, their highest blood alcohol levels occur after the binge. Other researchers (Matyas, C.; et al. , Am. J. Physiol. Heart Circ. Physiol. 2016, H1658–H1670)  recommend to allow at least 9 hrs after the binge before performing any hemodynamic or cardiac functional assessments to avoid the acute intoxication effects of the alcohol.  In addition, we have included additional information regarding the blood alcohol levels achieved using this alcohol feeding model by the diet and the binge dosing (Section 2.2).

  • Anesthesia, operator blinding, and bias control.
    Echocardiography under isoflurane without explicit blinding can bias functional readouts. Please report: (i) whether image acquisition and analysis were blinded; (ii) isoflurane concentration/timing; (iii) whether depth was matched across groups; and (iv) inter- and intra-operator variability or QC checks.

For the echocardiograpahy, we had a single investigator performing the measurements that was not blinded.  The isoflurane was set at 1.5% during the echo procedure and heart rate was monitored throughout.  Depth of anesthesia was gauged by heart rate and heart rates accepted for analysis were between 400-500 BPM. 

For cardiac catheterization, we had a single investigator performing the surgery and measurements, and that person was blinded to the study groups.

  • Statistical analysis inflates type-I error.
    Several figures appear to use two-way ANOVA followed by multiple post-hoc tests and additional groupwise t-tests highlighted on plots. This “double testing” increases false positives. Pre-specify a single analysis path, control FDR across families of related endpoints, and report effect sizes with 95% CIs.

This was a mistake on our part, as we intended to also show a stand-alone analysis of the female groups.  All statistical analyses were recalculated and we have reformatted the graphs and statistical markers for clarity to denote main effects of Sex or Ethanol. We used a two-way ANOVA as the primary analysis, and if interaction was found, we followed with Tukey’s HSD test for post-hoc pairwise comparisons.  We have also included detailed statistical results in the Supplemental.

Figure Adjustments:

  • Remove standalone t-test symbols (e.g., stars or brackets that came from uncorrected pairwise t-tests). (DONE)
  • Keep only ANOVA + Tukey comparisons on the plots. (DONE)
  • Point readers to a supplementary table for detailed effect sizes and CIs (to avoid cluttering the figures). (We have included a comprehensive table of statistical results in the Supplemental).

  • Ambiguity in Seahorse data processing and QC.
    The Methods describe OCR values being “normalized to non-mitochondrial respiration” and wells with inadequate responses to ADP or FCCP being excluded. Standard practice is subtraction of non-mitochondrial OCR rather than ratio scaling.

Thank you for pointing this out. The statement in the Methods section was a typographic error, as we did not normalize OCR values to non-mitochondrial respiration. Instead, we followed standard practice by subtracting non-mitochondrial OCR (measured after antimycin A/rotenone injection) from all preceding values to obtain basal and substrate-stimulated mitochondrial respiration rates. We have corrected the Methods section accordingly to accurately reflect our data processing approach. Our calculations were done similarly to the following publications (PMID: 29860557, PMID: 21799747, PMID: 23806974).

  • Validation gaps for H₂O₂ assays on isolated mitochondria.H₂O₂: Amplex Red assays should include catalase controls (to verify signal identity), and presentation of substrate-dependent rates (e.g., pyruvate/malate vs palmitoylcarnitine) with/without SOD to confirm matrix origin. Please add these controls, raw calibration curves, and representative traces.

Thank you for this feedback. We used the Amplex™ Red Hydrogen Peroxide/Peroxidase Assay Kit (A22188), which detects extracellular H₂O₂ through a coupled reaction with horseradish peroxidase, producing a fluorescent resorufin signal. This assay is well-established for quantifying H₂O₂ release from isolated mitochondria.

Regarding the suggested controls:

  1. While we recognize the value of catalase-based validation as done in a kit like the Amplex Red Catalase Assay Kit (A22180), we did not include catalase controls in these experiments. However, the specificity of the Amplex Red assay for H₂O₂, when paired with horseradish peroxidase, is well-documented, and all assays were conducted under tightly controlled conditions to minimize non-specific fluorescence.
  2. We agree that substrate-dependent comparisons can offer valuable mechanistic insight. However, we feel that these experiments fall outside the scope of the specific research question addressed in this project, which is focused on investigating sex-specific vulnerability in alcohol-associated cardiomyopathy by assessing cardiac and mitochondrial function in a preclinical model of chronic and binge alcohol exposure using the Lieber-DeCarli liquid diet. As such, we included only the substrate conditions used for mitochondrial respiration assays to maintain consistency across all experimental groups and outcomes. We did not include assays across multiple substrate conditions.
  3. In addition, we have provided standard calibration curves and representative fluorescence traces in the supplemental materials to support quantification and validate assay performance.

  • Model characterization is incomplete (BAC and systemic exposure).
    Blood alcohol concentrations (BAC) at defined timepoints—especially around imaging and tissue harvest—are essential to distinguish acute vs cumulative effects and to benchmark the model against the literature. Please provide BAC (or breath levels) and body-weight/consumption trajectories per sex.

While we do not have BALs from the time of sacrifice in these mice, we have included BALs that the model achieves at midday (on diet alone) and following ethanol binge.  This information was added to the Methods section in the description of the alcohol feeding approach.  Also, we do not have the body weights collected over time.  On average the ethanol group mice consume ~12-15 mls of the liquid diet (two mice per cage). The control mice are pair-fed, so we adjust the control mice food volume based on the average consumption of liquid diet for the ethanol mice.

  • Female hormonal status not controlled/reported.
    Given the sex-difference emphasis and estrogen’s known cardioprotective roles, estrous staging or hormone profiling is needed. At minimum, report cycle staging at testing or provide ovariectomy ± estradiol replacement data to test estrogen dependence.

The samples required for staging were not collected, so we are unable to provide these data.  The female mice did not exhibit significant variation within treatment groups and we found that they were cardioprotected against the effects of the ethanol chronic+binge feeding. We have included this information in the discussion of limitations, as well as future avenues of investigation to further assess the role of ovarian hormones.

  • ECM/fibrosis claims need orthogonal validation.
    Inferences about remodeling derived from mRNA (e.g., Col1a1/Col3a1 ratios, LARP6) should be corroborated at the protein and tissue level (Picrosirius Red or Masson trichrome, hydroxyproline, collagen I/III and TGF-β signaling proteins, or biaxial mechanics if stiffness is claimed).

We have included additional data in the revised manuscript.  Protein assessment of Collagen I and III and alpha-smooth muscle actin by western blot are presented.  We did not have fixed heart tissue to perform PSR staining in these groups.

Minor revision

  • Standardize nomenclature (e.g., “EtOH,” “HFrEF” if referenced, gene and protein formatting).
  • Verify reference formatting and ensure all in-text claims are supported by current literature.
  • Improve figure readability: ensure axis labels include units, avoid overplotting, and present mean ± SD/SEM consistently with superimposed datapoints.
  • Correct typographical errors and ensure consistent use of symbols (e.g., μ vs u).

We have made the requested corrections.

Reviewer 2 Report

Comments and Suggestions for Authors

Chronic alcohol misuse is the leading cause of non-ischemic dilated cardiomyopathy. Yet, the molecular mechanisms underlying this disease, as well as those underlying sex-specific differences in the development of alcohol-associated cardiomyopathy (ACM), are relatively underexplored. In this study, Edavettal et al. sought to investigate whether differences in mitochondrial function underlie sex differences in cardiac function induced by chronic+binge alcohol administration. The results are generally well presented, however, I have several concerns.

1. Line 310, repeat of “decrease in”

2. There is no information in the methods regarding whether normality of the data was assessed. Details should be included in the methods and, if normality testing reveals that the data are not normally distributed, non-parametric tests should be employed to evaluate statistical significance rather than parametric tests.

3. Histological assessment of the hearts should be included. Staining with quantitative analysis of fibrosis would provide orthogonal validation of increased fibrosis in males based on RT-qPCR results. Moreover, either 2 or 4 chamber views would provide additional morphological data to verify the echo findings and help resolve the apparent discrepancy between hypertrophy and unchanged HW in alcohol-fed female animals. 

5. Ejection fraction and fractional shortening can also be assessed by M-mode echocardiography. These data should be included as well, even if only in the supplement.

6. Lines 453-454, it is stated that “…LV mass was significantly reduced following chronic+binge alcohol exposure, consistent with pathological remodeling.” However, at least based on the echo data, hearts were not structurally altered. So what type of pathological remodeling are the authors referring to? Fibrosis?

Author Response

We appreciate the comments from the reviewers which have led us to include additional data and edits to many sections of the manuscript.  We believe that these edits have greatly improved the clarity and overall quality of the manuscript.

  1. Line 310, repeat of “decrease in”

Typo corrected.

  1. There is no information in the methods regarding whether normality of the data was assessed. Details should be included in the methods and, if normality testing reveals that the data are not normally distributed, non-parametric tests should be employed to evaluate statistical significance rather than parametric tests.

Prior to 2-way ANOVA analyses, data were assessed for normal distribution.  We have included a statement to this effect in the Methods under Statistical Analyses.

  1. Histological assessment of the hearts should be included. Staining with quantitative analysis of fibrosis would provide orthogonal validation of increased fibrosis in males based on RT-qPCR results. Moreover, either 2 or 4 chamber views would provide additional morphological data to verify the echo findings and help resolve the apparent discrepancy between hypertrophy and unchanged HW in alcohol-fed female animals. 

We have included additional data in the revised manuscript.  Protein assessment of Collagens I and III and alpha-smooth muscle actin by western blot are presented.  We did not have fixed heart tissue to perform PSR staining in these groups.  Another  reviewer pointed out that our statistical approach was flawed (i.e., using t-test to compare groups after ANOVA), which led us to reassess all of our statistics for the presented data strictly by 2-way ANOVA with Tukey's post-hoc.  This led to some changes in primarily the echo results where we no longer had significance in some of the wall and chamber dimensions. 

  1. Ejection fraction and fractional shortening can also be assessed by M-mode echocardiography. These data should be included as well, even if only in the supplement.

We have included these in the supplement.  We found no main effect of alcohol or sex.  We present the catheterization data as our primary functional endpoint, as we find this method to be very sensitive, repeatable and robust assessment of both systolic and diastolic cardiac function.

  1. Lines 453-454, it is stated that “…LV mass was significantly reduced following chronic+binge alcohol exposure, consistent with pathological remodeling.” However, at least based on the echo data, hearts were not structurally altered. So what type of pathological remodeling are the authors referring to? Fibrosis?

We have reworded for clarity, as to refer to this as pathological remodeling is an overstatement of our findings.

Reviewer 3 Report

Comments and Suggestions for Authors

Dear authors,

Your manuscript entitled “Female Cardioprotection in a Mouse Model of Alcohol-Associated Cardiomyopathy is Independent of Mitochondrial Sex Differences” addresses an important, yet under-investigated aspect of cardiovascular science, which concerns the contribution of biological sex to alcohol-induced susceptibility to cardiovascular disease via development of cardiomyopathy (ACM). In the present study, I believe that you have successfully designed and executed a cogent and thorough preclinical study, performed in male and female C57BL/6J mice, using a chronic+binge model of alcohol exposure. The combined use of several complementary techniques, including echocardiography, pressure-volume loop analysis, qPCR and high-resolution measurements of mitochondrial function, enables a comprehensive assessment of cardiac structural, functional, molecular and bioenergetic effects of alcohol.

The study provides strong evidence that female mice are relatively resistant to the toxic effects of chronic alcohol exposure to the heart, both in terms of impaired cardiac function in males, and despite comparable responses in mitochondrial respiration, ATP production, and ROS generation. This also leads the authors to suggest that the cardioprotective effects of female gender are secondary to a mechanism unrelated to the sex differences in mitochondrial bioenergetics.

Yet many parts in the manuscript need refinement and more detailed explanation. The following points present the main issues that require attention and improvement in my opinion:

Major Revisions

  1. The evidence that female cardioprotection is independent of mitochondrial differences is not fully supported by the data. Despite equal mitochondrial respiration and ATP/ROS indices between the sexes, males had a higher baseline OCR, indicating sexual distinctions in the mitochondria. Could you please explain how this observation is consistent with your conclusion and whether other pathways (e.g., calcium handling, mitochondrial dynamics) have been taken into account?
  2. Because sex hormones (in particular estrogen) are hypothesized to mediate the differences, direct assessment would enhance the manuscript (ie, measures of serum estradiol or manipulation such as ovariectomy or receptor blockade). Or at least include much more detailed discussion about that kind of limitation and propose future work which would potentially further clarify this point.
  3. According to the authors the manifestation of mitochondrial dysfunction is possibly not yet developed at this stage of disease. However, there is no longitudinal data or for earlier/later time points provided in this manuscript. It would be more appropriate in my opinion to perform a time-course experiment to further determine when mitochondrial damage began. If that is not possible then you should make it clear why you were prevented from doing so (for example for technical reasons) and further discuss and comment on this limitation.
  4. Mitochondrial measurements (i.e., ATP and ROS measurement) have evidence trends but without statistical significance. Please provide a power analysis to show that the number of samples are large enough to observe differences of biological significance or increase the n per group of the assays.
  5. The authors state that 30 days of exposure may indicate and represent early-stage disease, but fail to provide significate findings within the context of human ACM progression. Please enhance and expand the discussion on whether this pattern could be representive for clinical pathophysiology, particularly in women.
  6. The fact that female mice had an increased wall thickness on echocardiography (without a decrease in mass) is considered as adaptive. It was however not made clear why this is the case, or alternative explanations were not considered (e.g., subclinical remodelling). Could this be indicative of early hypertrophy or changed loading patterns?

Minor Revisions

  1. Perhaps the title should be rewritten to state clearly that the study is about early cardiomyopathy and that functional mitochondrial parameters remained intact. For instance: "Early Female Cardioprotection in a Mouse Model of Alcohol-Associated Cardiomyopathy Occurs Despite Preserved Mitochondrial Function"
  2. The phrase " cardiac dysfunction in ACM may precede or occur independently of mitochondrial impairment " would be safer to rewrite as "at this stage, cardiac dysfunction may occur without clear mitochondrial dysfunction".
  3. Please check the manuscript again and thoroughly for possible typographical errors, for example:

Line 308: duplicate phrase “decrease in decrease in LV + S mass”.

Line 622: “Dara are available…” should be corrected to “Data are available…”.

  1. Some figures (e.g., Fig 6, Fig 7) are essentially bar graphs with no single data point. You could also add scatter overlays to illustrate the distribution of the data and transparency in the statistical interpretation.
  2. Please double check for consistent reference formatting (certain references do not follow journal issue/page numbers or DOI spacing), as MDPI journals require uniform formatting across all references.
  3. This is a generally well-written manuscript that would however still benefit from minor language polishing for the sake of clarity and connectedness, especially in the Discussion part. For instance you could improve the sentence: “Interestingly, despite no observed differences…” by changing it into: “Interestingly, although no differences were observed…”
  4. For transparency and reproducibility, it would be helpful to include the raw data of experiments, the standard curves for measuring ATP level and ROS concentration, and the Ct values of qPCR analysis in the supplementary materials.

In summary, the manuscript adds to the field and will make an impact if revised to improve mechanistic, methodology and context clarity. There are some major and minor comments, which are provided above to help the authors in improving the manuscript.

Comments on the Quality of English Language

Minor improvements are required. See details in the review report.

Author Response

We appreciate the comments from the reviewers which have led us to include additional data and edits to many sections of the manuscript.  We believe that these edits have greatly improved the clarity and overall quality of the manuscript.

Major Revisions

  1. The evidence that female cardioprotection is independent of mitochondrial differences is not fully supported by the data. Despite equal mitochondrial respiration and ATP/ROS indices between the sexes, males had a higher baseline OCR, indicating sexual distinctions in the mitochondria. Could you please explain how this observation is consistent with your conclusion and whether other pathways (e.g., calcium handling, mitochondrial dynamics) have been taken into account?

After reviewing the comments, we realize that our title and some of our conclusions were quite overstated given our limited assessment of mitochondrial function by Seahorse. We have modified the title and edited the manuscript to temper our conclusions.  We have also expanded the limitations section of the discussion to include several points that the reviewers made in their comments. 

2. Because sex hormones (in particular estrogen) are hypothesized to mediate the differences, direct assessment would enhance the manuscript (ie, measures of serum estradiol or manipulation such as ovariectomy or receptor blockade). Or at least include much more detailed discussion about that kind of limitation and propose future work which would potentially further clarify this point.

Additional details were discussed in the limitations section about the role of ovarian hormones and mentioned as potential future avenue of investigation. 

3. According to the authors the manifestation of mitochondrial dysfunction is possibly not yet developed at this stage of disease. However, there is no longitudinal data or for earlier/later time points provided in this manuscript. It would be more appropriate in my opinion to perform a time-course experiment to further determine when mitochondrial damage began. If that is not possible then you should make it clear why you were prevented from doing so (for example for technical reasons) and further discuss and comment on this limitation.

We have edited the manuscript to include this in the limitations and as a potential future approach.

4. Mitochondrial measurements (i.e., ATP and ROS measurement) have evidence trends but without statistical significance. Please provide a power analysis to show that the number of samples are large enough to observe differences of biological significance or increase the n per group of the assays.

We performed a power analysis on the data that appeared closest to significance (Female ATP) and the number of samples needed to reach significance is 35 (p<0.05).

5. The authors state that 30 days of exposure may indicate and represent early-stage disease, but fail to provide significate findings within the context of human ACM progression. Please enhance and expand the discussion on whether this pattern could be representative for clinical pathophysiology, particularly in women.

We have included additional information in the Discussion about the limitations of the model that we are using.  The model does not produce significant cardiac fibrosis, which is a hallmark of clinical ACM, but the model is one of the few that produces significant cardiac dysfunction (cardiomyopathy).  

6. The fact that female mice had an increased wall thickness on echocardiography (without a decrease in mass) is considered as adaptive. It was however not made clear why this is the case, or alternative explanations were not considered (e.g., subclinical remodelling). Could this be indicative of early hypertrophy or changed loading patterns?

After our recalculation of all statistics based on our 2x2 design with 2-way ANOVA and Tukey's post hoc comparison, it appears that some of the changes in wall thickness are not statistically significant.  We have found in our other studies of severe chronic volume overload (no alcohol), that females had a greater hypertrophic capacity than males. In that study, the females were able to compensate for the stress of the volume overload and maintain normal function for a longer period than males.  It is not clear if that is happening in response to ethanol feeding, as our data do not consistently indicate hypertrophy in the females. 

Minor Revisions

  1. Perhaps the title should be rewritten to state clearly that the study is about early cardiomyopathy and that functional mitochondrial parameters remained intact. For instance: "Early Female Cardioprotection in a Mouse Model of Alcohol-Associated Cardiomyopathy Occurs Despite Preserved Mitochondrial Function"
  2. The phrase " cardiac dysfunction in ACM may precede or occur independently of mitochondrial impairment " would be safer to rewrite as "at this stage, cardiac dysfunction may occur without clear mitochondrial dysfunction".
  3. Please check the manuscript again and thoroughly for possible typographical errors, for example:

Line 308: duplicate phrase “decrease in decrease in LV + S mass”.

Line 622: “Dara are available…” should be corrected to “Data are available…”.

  1. Some figures (e.g., Fig 6, Fig 7) are essentially bar graphs with no single data point. You could also add scatter overlays to illustrate the distribution of the data and transparency in the statistical interpretation.
  2. Please double check for consistent reference formatting (certain references do not follow journal issue/page numbers or DOI spacing), as MDPI journals require uniform formatting across all references.
  3. This is a generally well-written manuscript that would however still benefit from minor language polishing for the sake of clarity and connectedness, especially in the Discussion part. For instance you could improve the sentence: “Interestingly, despite no observed differences…” by changing it into: “Interestingly, although no differences were observed…”
  4. For transparency and reproducibility, it would be helpful to include the raw data of experiments, the standard curves for measuring ATP level and ROS concentration, and the Ct values of qPCR analysis in the supplementary materials.

We have edited the manuscript in response to these comments.  In particular, we have toned down the overstatement of conclusions and removed speculation.  We also corrected the typographical errors and made the requested grammatical changes. We have updated the statistical analyses and presentation of all data, updated the references, and have included substantial raw data, standard curves, qPCR, WB, and table of 2-way ANOVA/Tukey's information in the supplemental materials. 

Round 2

Reviewer 1 Report

Comments and Suggestions for Authors

The study uses a chronic+binge (Lieber-DeCarli 5% v/v diet + two gavage “binges”) ethanol model in C57BL/6J mice (both sexes) over 30 days to test sex differences in ACM. Outcomes include echocardiography, invasive LV pressure–volume (PV) analysis, LV cytokine/fibrosis gene expression (RT-qPCR), collagen/α-SMA protein (immunoblot), and ex vivo mitochondrial assays (Seahorse OCR states, ATP luminescence, Amplex Red). Main findings: males develop systolic/diastolic dysfunction with pro-inflammatory/pro-fibrotic transcriptional shifts; females are largely protected; ex vivo mitochondrial respiration/ATP/ROS show sex differences (males higher OCR) but are not altered by ethanol.

Major comments
  1. The Discussion acknowledges no estrous staging or hormone measures. Given estrogenic cardio protection is posited, either (i) stage estrous, (ii) perform OVX ± E2 add-back, or (iii) time-block experiments and include serum E2/PG assays; at minimum, state estrous distribution at sacrifice.
  2. Risk of bias and experimental conduct: Echocardiography not blinded, and please clarify whether analysis of PV catheterization data was also blinded.

  3. OCR should be normalized to mitochondrial content/quality (e.g., citrate synthase activity, VDAC/TOM20 immunoblot, mtDNA copy number). Sex differences in OCR might simply reflect mitochondrial mass differences.
  4. Only malate/pyruvate are used. Consider adding palmitoyl-carnitine/ADP to test FAO capacity, relevant to sex bioenergetics; or explicitly justify Complex-I-only interrogation.
  5. Specify linear range, LOD/LOQ, intra/inter-assay CV, and demonstrate signal normalization to protein and mitochondrial markers. Provide raw standard curves in Supplement.
  6. mRNA–protein discordance. You report Col1a1/Col3a1 transcript shifts without protein/histology concordance after 30 d. To substantiate remodeling:

    • Include hydroxyproline or collagen area fraction; consider titin isoform or diastolic stiffness measures if fibrosis is minimal.
    • If fibrosis is truly absent at 30 d, temper claims and frame as pre-fibrotic transcriptional shifts. 

Minor points 
  • Standardize statistics notation (report exact P values).

  • Confirm whether tibial length was measured by one operator and whether the operator was blinded.

  • Ensure a single consistent abbreviation set in the Abbreviations list; check “SMC” vs “α-SMA” labeling in Figure 6 and legend consistency.

Author Response

Thank you for your thoughtful review. The manuscript has been significantly improved as a result of the reviewers' suggestions.

We have included a Word document with our responses.   

Reviewer 2 Report

Comments and Suggestions for Authors

The authors have done a great job addressing the previous critiques. However, I have a few further comments regarding the new data assessing the expression of collagen I and III.

  1. These results could be better described (Fig. 6).
  2. The reason multiple bands are shown for collagen I/III is not described. Are these different isoforms/modifications? Is there any band that is generally considered a positive signal either based on the company information for the specific antibody or the literature?
  3. How is the quantification being done for these blots? Are you just assessing the entire lane between a specific MW?
  4. For quantification, based on the methods the blots were stained for total protein. Total protein blots should be shown so readers/reviewers can verify the loading. 

Author Response

The authors have done a great job addressing the previous critiques. However, I have a few further comments regarding the new data assessing the expression of collagen I and III.

  1. These results could be better described (Fig. 6).
  2. The reason multiple bands are shown for collagen I/III is not described. Are these different isoforms/modifications? Is there any band that is generally considered a positive signal either based on the company information for the specific antibody or the literature?
  3. How is the quantification being done for these blots? Are you just assessing the entire lane between a specific MW?
  4. For quantification, based on the methods the blots were stained for total protein. Total protein blots should be shown so readers/reviewers can verify the loading. 

We appreciate the comments from the reviewer and realize that we needed to include additional details regarding the collagen western blots.  We have updated Figure 6 and associated legend. For collagen I and III, the three prominent bands between ~80 and 240 kDa were combined for quantification. Results were normalized to total protein of the lane. We have included the uncropped blots, total protein stain, and marked Regions of Interest for quantification in the Supplemental Materials (Figures S1-S3). We have also included a quantification of each individual band of interest for the collagen I and III blots, including full statistical analyses, in the Supplemental Materials.  We believe that this greatly improves the clarity, and readers may find the additional analyses useful for assessing distinct molecular weights for collagens I and III.

Thank you for helping us to improve our manuscript.

Reviewer 3 Report

Comments and Suggestions for Authors

The authors performed all necessary changes.

Author Response

Thank you for helping us improve the quality of our manuscript.